Molecular mechanism of point mutation-induced Monopolar spindle 1 (Mps1/TTK) inhibitor resistance revealed by a comprehensive molecular modeling study

Han Yan hanyan@wzhospital.cn yanhan˙zjwz@163.com 1
Wu Yungang 1
Xu Yi 2
Guo Wentao 3
Zhang Na 1
Wang Xiaoyi 1
1 Department of TCM Orthopedics & Traumatology, the First Affiliated Hospital of Wenzhou Medical University , Wenzhou , Zhejiang , China
2 Department of Pharmacy, the First Affiliated Hospital of Wenzhou Medical University , Wenzhou , Zhejiang , China
3 School of Pharmacy, Wenzhou Medical University , Wenzhou , Zhejiang , China
Silva Pedro
Electronic publication date: 2019 Jan 21
Publication date: 2019
Volume: 7
Electronic Location ID: e6299
Received 2018 Oct 11; Accepted 2018 Dec 18
Copyright: ©2019 Han et al.
Copyright year: 2019
Copyright holder: Han et al.
License: This is an open access article distributed under the terms of the Creative Commons Attribution License, which permits unrestricted use, distribution, reproduction and adaptation in any medium and for any purpose provided that it is properly attributed. For attribution, the original author(s), title, publication source (PeerJ) and either DOI or URL of the article must be cited.
License URL: https://creativecommons.org/licenses/by/4.0/

Keywords: Mutation, Monopolar spindle 1, Resistance mechanism, Molecular dynamics simulation

Funding: Wenzhou Science and Technology Plan Y20160380 Y20170156 This work was supported by grants from Wenzhou Science and Technology Plan (Y20160380 and Y20170156). The funders had no role in study design, data collection and analysis, decision to publish, or preparation of the manuscript.

==============================
Background

Monopolar spindle 1 (Mps1/TTK) is an apical dual-specificity protein kinase in the spindle assembly checkpoint (SAC) that guarantees accurate segregation of chromosomes during mitosis. High levels of Mps1 are found in various types of human malignancies, such as glioblastoma, osteosarcoma, hepatocellular carcinoma, and breast cancer. Several potent inhibitors of Mps1 exist, and exhibit promising activity in many cell cultures and xenograft models. However, resistance due to point mutations in the kinase domain of Mps1 limits the therapeutic effects of these inhibitors. Understanding the detailed resistance mechanism induced by Mps1 point mutations is therefore vital for the development of novel inhibitors against malignancies.

Methods

In this study, conventional molecular dynamics (MD) simulation and Gaussian accelerated MD (GaMD) simulation were performed to elucidate the resistance mechanisms of Cpd-5, a potent Mps1 inhibitor, induced by the four representative mutations I531M, I598F, C604Y, S611R.

Results

Our results from conventional MD simulation combined with structural analysis and free energy calculation indicated that the four mutations weaken the binding affinity of Cpd-5 and the major variations in structural were the conformational changes of the P-loop, A-loop and αC-helix. Energetic differences of per-residue between the WT system and the mutant systems indicated the mutations may allosterically regulate the conformational ensemble and the major variations were residues of Ile-663 and Gln-683, which located in the key loops of catalytic loop and A-loop, respectively. The large conformational and energetic differences were further supported by the GaMD simulations. Overall, these obtained molecular mechanisms will aid rational design of novel Mps1 inhibitors to combat inhibitor resistance.

Introduction

Mammalian cell division is accurately regulated by activation and inactivation of related proteins that manage progression through the phases of the cell cycle (Bertoli, Skotheim & De Bruin, 2013). To ensure segregation of duplicated chromosome during mitosis and meiosis, cell-cycle checkpoints have evolved that play essential roles in genome maintenance under varieties of stress conditions (Barnum & O’Connell, 2014). The spindle assembly checkpoint (SAC), one of cell-cycle checkpoints and a signaling cascade, prevents chromosome missegregation by delaying mitotic progression until all chromosomes are correctly attached to spindle microtubules (Hiruma et al., 2015). Inactivation of SAC leads to premature anaphase onset, and, therefore, chromosomal instability and aneuploidy, which is responsible for cell death or tumorigenesis (De Carcer, Perez de Castro & Malumbres, 2007; Marques et al., 2015; Salmela & Kallio, 2013).

The mono-polar spindle 1 (Mps1, also called TTK), one of the main components of SAC, is generally regarded as a master conductor of SAC signaling, which recruit in early mitosis to unattached kinetochores (Pachis & Kops, 2018). Mps1 has been proposed to be dysregulated in various cancer cells (Xie et al., 2017). For instance, mRNA expression of Mps1 is elevated in numerous of cancers relative to normal tissue. These include glioblastoma, osteosarcoma, hepatocellular carcinoma, breast cancer, and other cancers. Reduction in Mps1 levels or activity in these tumors can lead to loss of cell viability; therefore, inhibition of Mps1 has been regard as an attractive strategy to target cancers, especially those with chromosomal instability (Daniel et al., 2011; Xie et al., 2017). Many structurally diverse Mps1 inhibitors have been reported and have undergone preclinical assessments in recent years, such as NMS-P715, Cpd-5, CCT251455, MPS1-IN-3, reversine, MPI-0479605, AZ3146 (Colombo et al., 2010; Hewitt et al., 2010; Hiruma et al., 2017; Naud et al., 2013; Tannous et al., 2013; Tardif et al., 2011). Unfortunately, in cancer cells, the resistance to Mps1 inhibitors eventually occurs and this resistance is mostly due to mutations in the conserved ATP binding pocket of the Mps1 kinase domain, which leads to a remarkably attenuation of the therapeutic efficiency of the Mps1 inhibitors (Hiruma et al., 2017; Koch et al., 2016).

The kinase domain of Mps1 adopts a bi-lobal structure, a smaller N-terminal domain, and a larger C-terminal domain connected by kinase hinge, which together form the catalytic site to ensure the transfer of a phosphate from adenosine triphosphate (i.e., ATP) to a substrate hydroxyl (Fig. 1A). The N-terminal domain contains six stranded antiparallel β-sheets, an important regulatory αC-helix, and a phosphate binding loop (i.e., P-loop). The C-terminal domain consists of seven α-helices, two β-sheets, a catalytic loop, and an essential activation loop (i.e., A-loop, residues 676–685) (Chu et al., 2008; Wang et al., 2009). Recently, Koch et al. (2016) reported four point mutations (I531M, I598F, C604Y, S611R) in the kinase domain of Mps1 that gave rise to the Cpd-5 resistance, but retained wild type (WT) catalytic activity. Cpd-5 (N-(2,6-diethylphenyl)-8-((2-methoxy-4-(4-methylpiperazin-1-yl)phenyl)amino)-1-methyl-4,5-dihydro-1H-pyrazolo[4,3h] quinazoline-3 carboxamide), a derivative of NMS-P715, has been reported to display higher potency toward Mps1 than NMS-P715, reversine, MPI-0479605 (Koch et al., 2016). Recently, Chen et al. (2018) reported resistance mechanisms of inhibitors to Mps1 C604Y Mutation, however, it is not clear from the structures why other mutations (I531M, I598F, S611R) would cause resistance. Molecular dynamics (MD) simulations may be helpful to explain these resistance mutations by monitoring the dynamics of the protein and its interactions with bound inhibitors.

Figure 1 Overview of the kinase domain of Mps1 and point mutations.

(A) Overview the structure of the Mps1 kinase domain. The point mutations of I531M, I598F, C604Y, S611R are colored magenta balls. (B) Two-dimensional structure of Cpd-5.

In this study, the WT Mps1 (Mps1WT) and four mutants associated with Cpd-5 resistance, including I531M (Mps1I531M), I598F (Mps1I598F), C604Y (Mps1C604Y), S611R (Mps1S611R), were chosen to create an overall view of the structural dynamics. Herein, conventional molecular dynamics (MD) simulations combined with structural analysis and Molecular Mechanics/Generalized Born Surface Area (MM/GBSA) free energy calculations were employed to elucidate the impact of point mutations on the conformational and energetic difference between WT and mutant systems. In addition, Gaussian accelerated molecular dynamics (GaMD) simulations in conjunction with structural analysis, principal component analysis (PCA) and 2D free energy calculations were performed to sample more conformational space due to conventional MD simulation remains limited conformational ensembles. Overall, the comprehensive molecular modeling study performed in this study can be effective to study the resistance mechanism. More importantly, the insights from this study may have an essential application in designing novel Mps1 inhibitors to combat inhibitor resistance.

Materials & Methods

Construct the initial structures of Mps1WT, Mps1I531M, Mps1I598F, Mps1S611R

The three-dimensional structure of the Mps1C604Y/Cpd-5 complex was retrieved from Protein Data Bank (PDB) database (PDB entry: 5MRB) (Hiruma et al., 2017). Mps1WT, Mps1I531M, Mps1I598F, Mps1S611R were constructed by using the EasyModeller program by substituting specific residues (Fiser & Sali, 2003; Kuntal, Aparoy & Reddanna, 2010). Then, the modeled structures were refined by Chimera software, including modelling the A-loop, adding missing side-chains and missing hydrogen atoms (Pettersen et al., 2004).

Conventional molecular dynamics (MD) simulations

The crystal structure of Mps1C604Y/Cpd-5, and the modeled structures of Mps1WT/Cpd-5, Mps1I531M/Cpd-5, Mps1I598F/Cpd-5, Mps1S611R/Cpd-5 were used as the initial structures for the conventional MD simulations. Prior to conventional MD simulations, the force field parameters for the proteins and ligands were generated using LEaP module in Amber 16 software with AMBER ff14SB force field and general Amber force field 2 (GAFF2). Thereafter, each system was immersed in a water box by the TIP3P water model with at least a 12 Å distance around the protein-ligand complex. Finally, an appropriate number of counter ions were added to keep the electroneutrality.

Before each productive MD simulation, a sophisticated protocol was applied, including minimization, heating and equilibration. The first step of energy minimization was to employed a harmonic restraint of 5 kcal mol−1 Å−2 to the solute and side chains of protein. The second step was to allow all atoms to move freely without any restraint. In each step, minimization was performed using the steepest descent algorithm for the first 5,000 steps and the conjugated gradient algorithm for the succeeding 5,000 steps. Afterwards, each system was heated to 310 K in the NVT ensemble using a Langevin thermostat with harmonic restraints of 4 kcal mol−1 Å−2. Then, 1ns unconstrained NPT dynamics at 310 K and 1 bar was applied to equilibrate each system. Lastly, 240 ns productive conventional MD simulations were carried out under the NPT condition. During the simulations, Periodic boundary condition, Langevin temperature scaling, Particle Mesh Ewald (PME) algorithm with cutoff 8.0 Å and SHAKE algorithm was applied (Essmann et al., 1995; Izaguirre et al., 2001; Kräutler, Van Gunsteren & Hünenberger, 2001). A time step of 2 fs was performed and coordinates were recorded every 2 ps for further analysis.

End-point free energy calculations

The binding free energy (ΔGcalc) of Cpd-5 binding to Mps1WT, Mps1I531M, Mps1I598F, Mps1C604Y, Mps1S611R was calculated by using the end-point molecular mechanics generalized Born surface area (MM/GBSA). The MM/GBSA approach has been extensively used in understanding the mechanisms of mutation-induced drug resistance (He et al., 2018; Liu et al., 2017; Pan et al., 2013). The ΔGcalc was computed by using the following equations:

(1) ΔGcalc=ΔEMM+ΔGsol−TΔS

(2) ΔEMM=ΔEint+ΔEvdW+ΔEelec

(3) ΔGsol=ΔGGB+ΔGSA

(4) ΔGSA=γΔA+b

where ΔEMM, ΔGsol, and −TΔS represent the molecular mechanics interaction energy, solvation energy, and entropy term (Eq. 1). In Eq. 2, the ΔEMM consists of the change of internal energy (ΔEint), van der Waals energy (ΔEvdW), and electrostatic energy (ΔEelec). In Eq. 3, the ΔGsol contains the polar part (ΔGGB) and the nonpolar part of the desolvation energy (ΔGSA). In this study, ΔEint was canceled by using a single trajectory strategy in order to reduce the noise (Sun et al., 2013). The ΔGGB was calculated using a GBOBC1 model (igb = 2) (Onufriev, Bashford & David, 2000). The ΔGSA was calculated by Eq. 4. Where ΔA is the change of the solvent accessible surface area (SASA), and the fitting coefficients γ and b were set to 0.0072 kcal mol−1Å−2 and 0, respectively. Simulation trajectories from 200 to 240 ns with 200 snapshots were employed to binding free energy calculations and free energy decompositions.

Gaussian accelerated molecular dynamics (GaMD) simulation

The equilibrated structures extracted from conventional MD simulation trajectories were chosen as the initial structure for the GaMD simulations. During GaMD simulations, a non-negative harmonic boost potential is added to smooth the studied system potential energy surface in order to decrease the energy barriers and accelerate the conformational sampling (Miao & McCammon, 2016a; Miao & McCammon, 2016b). When the system potential V (r→) is lower than a threshold energy (E), a harmonic boost potential V∗r→ is added as Eq. (5). k is the harmonic force constant. When the system potential is above the E, the boost potential is set to zero as Eq. (6).

(5) V∗r→=Vr→+12kE−Vr→2Vr→<E

(6) V∗r→=Vr→Vr→≥E

GaMD simulation provides the total potential boost, dihedral potential boost, and dual potential boost in order to accelerate the MD simulations. Herein, dual potential boost was applied to the GaMD simulations. The dual potential boost parameters were computed from an initial ∼4 ns NVT conventional MD simulation without any potential boost. Afterwards, 1 ns GaMD simulation were performed, in which the potential boost was updated every 100 ps to reach equilibrium values. Eventually, 400 ns GaMD simulation for each system was submitted in the NVT ensemble. During the simulations, the temperature was regulated using a Langein thermostat. PME algorithm and the non-bonded with cutoff 10.0 Å to consider the long-range electrostatic interactions and non-bonding interactions (Essmann et al., 1995). Atomic coordinate trajectory was recorded every 2 ps.

Principal component analysis (PCA) and free energy calculations for the GaMD simulations

PCA calculations contain diagonalization of the covariance matrix of positional deviations among a structural ensemble. The structures from MD simulation trajectories were aligned to remove the translational and rotational motions. PCA was performed for the trajectories of 400 ns GaMD simulations by using the Bio3D package of R (Skjaerven et al., 2014). Thereafter, the potential boost combined with the principal component 1 (PC1) and principal component 2 (PC2) calculated from PCA were applied to recover the free energy map by cumulant expansion to the 2nd order method (Miao et al., 2014; Roe & Cheatham 3rd, 2013).

Results and Discussion

The overall structural properties

The structural dynamics of Cpd-5 bound with WT Mps1 or mutant Mps1 were analyzed by performing 240 ns conventional MD simulations. To qualitatively investigate the stability and overall convergence of the simulated systems, the root mean square deviation (RMSD) of protein backbone atoms, and the heavy atoms of the Cpd-5 were calculated and plotted against time in Figs. 2 and 3. As plotted in Fig. 2, all the RMSD values of backbone atoms of Mps1 have a small fluctuation after 50–170 ns, indicating all the systems achieve equilibrium at ∼50–170 ns. The RMSD of the heavy atoms of the Cpd-5 in each system maintained relative stable (RMSD < 2 Å) during the 240 ns simulation (Fig. 3). As expected, the average RMSD values of both the Mps1 protein and the Cpd-5 follow the order of WT systems <mutant systems (Figs. 2 and 3). These findings are directly consistent with the experimental data that Cpd-5 is more stable in WT Mps1 than in the mutants. Thereafter, the trajectories of the last 60 ns simulations were extracted for the following structural and energetic analysis.

Figure 2 Time evolution of the RMSD values of backbone atoms of the Mps1 protein in the five studied systems from conventional MD simulations.

The values reflect the equilibration of each protein relative to the initial structures.

Then, the root-mean-square fluctuations (RMSFs) of the protein backbone, which represents the flexibility and mobility of protein backbone, were calculated and averaged for all the simulated systems to evaluate the local structure transformations in more detail. As shown in Fig. 4, all the simulated systems share similar RMSF distributions and similar trends of dynamics features. However, we noticed the P-loop region of Mps1I531M, Mps1I598F, Mps1C604Y, the αC-helix region of Mps1I598F, the A-loop region of all the four mutant systems, which exhibited amplified fluctuations when these regions were compared to WT system. The alignment of the last snapshot between WT (yellow) system and mutant (cyan) systems showed a highly similar pattern with minor adjustments in the regions of P-loop, αC-helix, A-loop (Fig. 5). For instance, compared with Mps1WT, the P-loop region of Mps1I531M, Mps1I598F, Mps1C604Y and the αC-helix region of Mps1I598F change into upward-moving conformation compared with Mps1WT (Figs. 5B–5D). The A-loop region of mutant systems showed an outward-moving conformation (Figs. 5A–5D). This finding was also consistent with the previous study that when Mps1 binds with Cpd-5, the A-loop region of Mps1C604Y changes into an outward-moving conformation, but not the Mps1WT (Chen et al., 2018). Compared with the previous results, we also found that the P-loop had a slight conformational change (Chen et al., 2018), particularly in Mps1I598F and Mps1C604Y systems. Overall, these results indicated that mutation-induced conformational change might be the main driving force for the redistributed energies. Hence, the MM/GBSA approach was employed to determine the key molecular determinants.

Figure 3 Time evolution of the RMSD values of cpd-5 in the five studied systems from conventional MD simulations.

The values reflect the equilibration of Cpd-5 in different protein relative to the initial structures.

Figure 4 RMSFs of backbone atoms versus residue number in the five studied systems from conventional MD simulations.

RMSF provides a picture of overall movement of a residue within a reference frame. A higher RMSF value represents a larger conformational change in spec.

Figure 5 Alignment of the last snapshot between Mps1WT/cpd-5 and the mutant systems from conventional MD simulations.

(A) Mps1WT/cpd-5 (cyan) and Mps1I531M/Cpd-5 (yellow). (B) Mps1WT/cpd-5 (cyan) and Mps1I598F/Cpd-5 (yellow). (C) Mps1WT/cpd-5 (cyan) and Mps1C604Y/Cpd-5 (yellow). (D) Mps1WT/cpd-5 (cyan) and Mps1S611R/Cpd-5 (yellow).

Figure 6 Differences of each residue contribution between the WT system and the mutant systems.

(A) The energetic differences between the WT system and the mutant systems of each residue contribution to Cpd-5 binding; (B) schematic view of the key residues; (C) detailed view of the key residues; (C) detailed view of the key residues.

Binding free energy and decomposition analysis

As an important complement to the binding mode and structural analysis discussed above, the binding free energies of the Cpd-5 to the WT system and the mutant systems calculated from the MM-GBSA approach were summarized in Table 1. According to Table 1, the predicted binding free energies (ΔGcalc) of Cpd-5 to Mps1WT, Mps1I531M, Mps1I598F, Mps1C604Y, Mps1S611R were −59.73 ± 4.72, −51.89 ± 4.14, −52.00 ± 4.61, −49.77 ±3.67, and −54.66 ± 3.74 kcal/mol, respectively. It can be observed that the ΔGcalc follow the order of WT systems <mutant systems, that is to say, the ΔGcalc show a high correlation with the reported experimental data of IC50. Then, individual components of binding free energy were analyzed, which could provide useful information to understand the inhibitor resistance mechanism followed by individual contribution decompositions to the binding free energies. As shown in Table 1, the van der Waals contributions of Cpd-5 to Mps1WT, Mps1I531M, Mps1I598F, Mps1C604Y, Mps1S611R were −73.57 ± 4.62, −62.15 ± 3.78, −63.28 ± 5.09, −61.16 ±3.89, −66.43 ± 4.28 kcal/mol, respectively, which determine the difference of the binding free energies between WT and mutant systems.

Table 1 Binding free energies of Cpd-5 in WT and mutant systems (kcal/mol).

Name	ΔEvdW	ΔEelec	ΔGGB	ΔGSA	ΔGcalc	IC50 (nM)	
Mps1WT	−73.57 ± 4.62	−26.98 ± 7.14	49.74 ± 5.95	−8.92 ± 0.48	−59.73 ± 4.72	33.2	
Mps1I531M	−62.15 ± 3.78	−27.71 ± 7.11	47.89 ± 5.56	−7.91 ± 0.41	−51.89 ± 4.14	119.0	
Mps1I598F	−63.28 ± 5.09	−28.04 ± 7.15	47.37 ± 5.66	−7.72 ± 0.55	−52.00 ± 4.61	72	
Mps1C604Y	−61.16 ± 3.89	−30.23 ± 5.44	49.61 ± 4.63	−7.98 ± 0.36	−49.77 ± 3.67	374.8	
Mps1S611R	−66.43 ± 4.28	−27.79 ± 6.75	47.75 ± 4.51	−8.18 ± 0.42	−54.66 ± 3.74	129.9	
Notes.

ΔEvdW Van der Waals energy

ΔEele electrostatic energy

ΔGGB electrostatic contribution to solvation

ΔGSA non-polar contribution to solvation

ΔGcalc binding free energy

IC50 half maximal inhibitory concentration

To further probe the interaction between Cpd-5 and Mps1, per-residue decomposition analysis was conducted to obtain a more detailed description of each residue contribution to the binding free energy by MM/GBSA decomposition approach. Energetic differences of per-residue between the WT system and the mutant systems (ΔΔG = ΔGWT − ΔGmutant) were plotted to highlight the key residues, which induced redistribution of binding free energy. As shown in Fig. 6, the negative values indicated that the residues of the WT Mps1 protein form stronger interactions with Cpd-5 than the mutant Mps1 protein, while the positive values suggested that the residues of the mutant Mps1 protein form stronger interactions with Cpd-5 than the WT Mps1 protein. As plotted in Fig. 6A, the key residues of Ile-663 and Gln-683 have significant stronger interactions to Cpd-5 in the WT system than in all the mutant systems. Notably, the residues of Ile-663 and Gln-683 are located in the two flexible key loops of catalytic loop and A-loop, respectively, which are important in rotating the Asp-Phe-Gly (DFG) motif into proper orientation for catalysis and substrate binding (Figs. 6B and 6C). These observations may be helpful to aid rational design of novel Mps1 inhibitors to overcome inhibitor resistance. For instance, moderately increasing the hydrophobicity of the compound to interact with Ile-663 to stabilize the catalytic loop and forming hydrogen bonds with Gln-683 to stabilize the A-loop.

Figure 7 Time evolution of the RMSD values of backbone atoms of the Mps1 protein in the five studied systems from GaMD simulations.

The values reflect the equilibration of each protein relative to the initial structures.

Figure 8 Time evolution of the RMSD values of Cpd-5 in the five studied systems from GaMD simulations.

The values reflect the equilibration of Cpd-5 in different protein relative to the initial structures.

Figure 9 Alignment of the last snapshot between Mps1WT/cpd-5 and the mutant systems from GaMD simulations.

(A) Mps1WT/cpd-5 (lime) and Mps1I531M/Cpd-5 (magenta). (B) Mps1WT/cpd-5 (lime) and Mps1I598F/Cpd-5 (magenta). (C) Mps1WT/cpd-5 (lime) and Mps1C604Y/Cpd-5 (magenta). (D) Mps1WT/cpd-5 (lime) and Mps1S611R/Cpd-5 (magenta).

Figure 10 PCA scatter plot of 50,000 snapshots from GaMD simulations along the first two principal components and plotted against time.

(A) Mps1WT/cpd-5. (B) Mps1I531M/Cpd-5 (magenta). (C) Mps1I598F/Cpd-5. (D) Mps1C604Y/Cpd-5; (E) Mps1S611R/Cpd-5.

GaMD simulations

All-atom conventional MD simulations are still limited to the conformational ensembles because of the possible energy barriers between various intermediate states. Therefore, an enhanced sampling method, which can take samples at various intermediate states is required. The traditional enhanced sampling often requires predefined parameters, such as root-mean-square distance atom distances, torsional dihedral, which usually needs expert knowledge of the studied systems. However, the enhanced sampling technique of GaMD simulation avoids such a requirement. In this study, we applied the GaMD simulation technique to further explore more conformational states. Following the initial 240 ns conventional MD simulation, the last snapshot of each system was used as the initial structure to GaMD simulation.

The RMSDs of the protein backbone and the heavy atoms of Cpd-5 with respect to the starting conformations were calculated after alignment of all the conformations from GaMD simulation by removing the rotational and translational motions (Figs. 7 and 8). As shown in Fig. 7, the RMSD of protein backbones of all the systems achieved equilibrium after 30–200 ns GaMD simulations. Figure 8 showed the RMSDs of the heavy atoms of Cpd-5 in each system maintained dynamic constant during the 400 ns of GaMD simulation, indicating the stability of the Cpd-5 in the studied systems.

In addition, the average RMSD values of both the protein and the ligand follow the order of WT systems <mutant systems, which are similar with the results from the conventional MD simulation. The above observations suggested that these four point mutations allowed larger conformational changes and more variability among protein subunits. Then, the last snapshots of each mutant system (magenta) extracted from GaMD simulation trajectories were superimposed with the last snapshot of WT system (green) to obtain an intuitionistic description of the conformational changes among these systems. As shown in Figs. 9A–9D, most conformational changes occurred in the P-loop and A-loop. These findings also supported the observation from conventional MD simulations. Compared with Mps1WT, the P-loop of Mps1I598F, Mps1C604Y, and A-loop of all the mutant systems changed into upward-moving conformations and outward-moving conformations, respectively.

PCA was performed to further characterize the conformational transitions and plotted against time (Fig. 10). PCA reduces the dimensionality of large data sets and the vectors with the highest eigenvalues represent the most significant principal components (PCs). When principal components are plotted against each other, similar structures are clustered. Theoretically, each cluster shows a different protein conformational state. As shown in Fig. 10 and Fig. S1, the conformational states of both the WT and mutant systems were dynamics and functions during 0–400 ns GaMD simulations, and finally, stabilized in one state. Compared to the WT system, the mutant systems exhibited more structural clusters and larger eigenvalues. These conformer plots highlighted the major differences between WT and mutant systems, which interpreted the observations of large conformational changes when resistance mutations occurred by conventional MD simulations and GaMD simulations.

Figure 11 Two-dimensional free energy landscape of the first and second principal components calculated from GaMD simulations.

(A) Mps1WT/cpd-5. (B) Mps1I531M/Cpd-5 (magenta). (C) Mps1I598F/Cpd-5. (D) Mps1C604Y/Cpd-5. (E) Mps1S611R/Cpd-5.

The free energy landscape was utilized to further explain the relationship between the conformational change and energetic change. More energetic wells (dark blue regions) represent the protein underwent larger conformational change. As shown in Fig. 11, the WT system was confined to a major energetic deep well throughout the simulation (Fig. 11A), while numerous of energetic deep wells were observed for the four mutant systems (Figs. 11B–11F), highlighting an ensemble of different conformational states distributed over a large free energy space. Additionally, the free energy landscape between the four mutant systems also showed significant difference (Figs. 11B–11F), suggesting the four mutations may allosterically regulate the conformational ensemble to induce inhibitors resistance.

Conclusions

In this study, molecular modeling, conventional MD simulations and GaMD simulations successfully clarified the resistance mechanism induced by the four point mutations of I531M, I598F, C604Y, S611R, both structurally and energetically. The binding free energies of Cpd-5 to the WT system and mutant systems were well predicted and the decomposition of the individual energy terms suggested the major variation of Cpd-5 between the WT system and the mutant systems were van der Waals interactions. Structural analysis revealed the conformational changes of the P-loop, A-loop and αC-helix play crucial role for Cpd-5 resistance. Further energetic differences of per-residue between the WT system and the mutant systems revealed the four mutations may allosterically regulate the conformational ensemble and the major variations were residues of Ile-663 and Gln-683, which were located in the key loops of catalytic loop and A-loop, respectively. In addition, GaMD simulations supported the observations from conventional MD simulations. PCA results and free energy landscape from GaMD simulations indicated Mps1 underwent large conformational changes when the resistance mutations occurred. In summary, our study not only revealed the resistance determinants of Cpd-5 to the four point mutations, but also provides some valuable information for structure based design of novel inhibitors of Mps1 in the future.

Supplemental Information

Data S1 Raw data corresponding to Figures 1–11 and Figure S1

Click here for additional data file.

Supplemental Information 1 Alignment of PCA scatter plot of the four mutant systems along the first two principal components

Click here for additional data file.

Additional Information and Declarations

Competing Interests

Author Contributions

Data Availability

The authors declare there are no competing interests.

Yan Han conceived and designed the experiments, performed the experiments, approved the final draft.

Yungang Wu conceived and designed the experiments, performed the experiments, analyzed the data, contributed reagents/materials/analysis tools, prepared figures and/or tables, authored or reviewed drafts of the paper, approved the final draft.

Yi Xu conceived and designed the experiments, performed the experiments, analyzed the data, contributed reagents/materials/analysis tools, prepared figures and/or tables.

Wentao Guo performed the experiments, analyzed the data, contributed reagents/materials/analysis tools.

Na Zhang performed the experiments, authored or reviewed drafts of the paper.

Xiaoyi Wang conceived and designed the experiments, authored or reviewed drafts of the paper.

The following information was supplied regarding data availability:

Raw data is available in the Supplemental Files.

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
