# Peer review of "Molecular mechanism of point mutation-induced Monopolar spindle 1 (Mps1/TTK) inhibitor resistance revealed by a comprehensive molecular modeling study"

_PeerJ, doi:10.7717/peerj.6299_

## Round 0.1 · original submission · Minor Revisions

Our reviewers consider your paper well-written and clear. You should, however, compare your results for C604Y with the (so far uncited) study of Chen et al. doi:10.3390/molecules23061488. Please address also the other minor points remarked by the reviewers.

Reviewer 1 ·

Basic reporting

no comment

Experimental design

no comment

Validity of the findings

no comment

Additional comments

This is an interesting report on the molecular dynamics (MD) simulation analysis of the structure of Mps1 and four resistant mutants. To my understanding the conclusions are in general valid and they are well presented and illustrated. Some minor modifications would need to be made before it can be published.
1. Which method used in this study to reweight the free energy map from GaMD simulation? Maclaurin series to the 10th order, cumulant expansion to the 2nd order or other? The authors should state in the method section.
2. Generally, please use the term "inhibitor" and not "drug" due to the target of Mps1 is a very attractive potential target, but no clinically proven drugs.
3. Please add discussion to propose how to used MD simulation results to discover new inhibitors to combat inhibitor resistance.

·

Basic reporting

In this study, Han et al. report on their findings of molecular dynamics (MD) and Gaussian accelerated MD (GAMD) simulations of Mps1/TTK wildtype and mutant versions together with the small molecule inhibitor Cpd-5.
By using informatic tools, such as MD simulations and structural analysis as well as GaMD they show that the point mutations in Mps1 induce changes in the structure of the kinase domain of Mps1, namely the P-loop and αC-helix, which they propose to possibly explain why the mutants bind less stably to Cpd-5. The major change identified was in the van der Waals interactions between Cpd-5 and the mutants compared to Mps1-WT. The authors also identified the residues Ile-664 and Gln-683, located in essential loops of the Mps1 kinase, that contribute majorly for the differences in binding seen between the WT and the mutants to Cpd-5.
1. BASIC REPORTING
- The paper is in most parts well written and clear. However, in specific cases (for example in the introduction) some formulations are misleading and need clarification -> for details see comments under Point 4.
- The overall figure quality is good. I feel however that the figures 2 and 3 are difficult to read because of the overlapping curves (and in fig 2 the curves overlap with the labelling). Maybe splitting would be a better option (or show individual plots in the supplement) The figure legends would need some improvement and a few more explanations of what is depicted.
- The introduction and cited literature show the context in a comprehensive and nice way. I would have only a few suggestions -> for details see comments under Point 4.
- One thing that is completely missing is the citation and mentioning of a study that basically shows similar results and in my view was followed up by the authors to produce this manuscript (experimental methods, study design and style of manuscript). “Chen Y, Yu W, Jiang C-C, Zheng J-G. Insights into Resistance Mechanisms of Inhibitors to Mps1 C604Y Mutation via a Comprehensive Molecular Modeling Study. Molecules 2018; 23. doi:10.3390/molecules23061488”. The authors should not try to ‘hide’ this fact but use it to their advantage and elaborate on why they went a step further (other mutations than C604Y) and included new findings.

Experimental design

- The experimental design is fine. Since my expertise lies more in the biological side of small molecule inhibitor research, I can however not fully judge the exact details of the performed methods and techniques the authors use here.

Validity of the findings

- The authors should compare their results for C604Y with the previous study I mentioned above (Chen et al., Molecules 2018). Can their results and measurements explain why C604Y is inhibited by Cpd-5 but not Reversine?
- The authors claim in the abstract that ‘Overall, these obtained molecular mechanisms will aid rational design of novel Mps1 inhibitors to combat drug resistance.’ I would be interested if this method can really do that and I know this might be very difficult to prove but what happens for example if one uses a structure of a small molecule inhibitor which has not been tested in these mutants yet. Or at least explain in more detail how this method can be used for these predictions (for an audience like me).

Additional comments

Mps1 is the name of the yeast version of this kinase. I would actually prefer if the authors would relate to it as Mps1/TTK (at least in the title).
The authors use the word ‘systems’ when talking about the different Mps1 versions. Maybe ‘protein’ would be a better word.
Minor points:
- The English language should be improved. Some points are confusing and the way they are phrased is wrong.
i. Example on line 60-62: It sounds like Mps1 recruits unattached kinetochores which is not true, but: Mps1 is recruited in early mitosis to unattached kinetochores.
ii. Example on line 71-74: Phrase is not well structured, and it is not understandable to a broader audience. I guess what the authors mean is that in cancer cells, the resistance to Mps1 inhibitors eventually occurs and this resistance is mostly due to mutations/changes in the ATP binding pocket of the Mps1 kinase domain which leads to a remarkably attenuation of the therapeutic efficiency of the Mps1 inhibitors.
iii. Line 76: change to ‘which together form the catalytic site to ensure the transfer of a phosphate from...’ and not transformation.
iv. On several occasions (e.g. line 186) Mps1 is written Msp1
v. Decide on using one- or three letter code for amino acids (one letter code - e.g. I598F vs. three-letter Ile-664)
vi. Line 240 – line 242: This first part of the sentence makes no sense and needs revision.
vii. Line 245: include ‘is/are’ in front of ‘still limited’?
viii. Line 262: include ‘are’ in front of ‘similar’
ix. Line 283 – 290: this part needs a revision. I do not understand most of the results because I have the feeling words are used wrong (e.g. line 289 ‘different’ -> difference?)
x. Line 313: what are ‘challenging’ conflicts of interest?

---

## Round 0.2 · accepted · Accept

Thank you for addressing the remaining issues. I am glad to approve your paper for publication.

There are still a few minor English tweaks needed ("These observations may helpful " instead of "These observations may be helpful " ) .
You should check those during production

# Reviewer 1 ·

Basic reporting

No comment

Experimental design

No comment

Validity of the findings

No comment

Additional comments

Sufficiently addressed my previous comments.

·

Basic reporting

All points were addressed.

Experimental design

See my initial review.

Validity of the findings

See my initial review.

Additional comments

Thank you for addressing our comments and revising your manuscript.